# Optimization of *Hermetia illucens* (L.) egg laying under different nutrition and light conditions

**Laura I. Macavei**[1]*, **Giacomo Benassi**[1¤], **Vlad Stoian**[2], **Lara Maistrello**[1]

**1** Department of Life Science, University of Modena and Reggio Emilia, Emilia-Romagna, Italy,
**2** Department of Microbiology, Faculty of Agriculture, University of Agricultural Sciences and Veterinary Medicine Cluj-Napoca, Cluj-Napoca, Cluj, Romania

¤ Current address: Kour Energy SRL, Reggio Emilia, Emilia-Romagna, Itay
* lmacavei@unimore.it

**Data Availability Statement:** All relevant data are within the manuscript and its Supporting Information files.

**Funding:** This work was partially supported by the project VALORIBIO – ERDF Emilia-Romagna

## Abstract

The black soldier fly *Hermetia illucens* is gaining growing interest as a tool for the valorisation of bio-waste in a circular economy perspective. Although a wide variety of studies are available for larvae rearing, the indoor breeding of adults still presents a great challenge for industrial purposes. This study was designed to assess the simultaneous influence of 3 different light sources (the Mix of LED UV:blue:green 1:1:3, White LED, Neon light) and 3 types of nutrition (sugar and water, only water, no sugar no water) on adult performances, obtaining different egg production parameters that included the number and weight of the egg masses and single eggs laid by the females, the duration of pre-oviposition and oviposition period, the adult life span, the hatchability of the eggs. Our results showed that production parameters are influenced mainly by nutrition rather than light factor, although light plays an important secondary role. Moreover, the presence of sugar positively affects the egg production (12,93–27,10 mg eggs/female) and increases oviposition period (18,2–31,8 days) and adult lifespan (20,79–27,11 days). Light sources also affect egg production parameters, with the exposure to the Mix of LEDs resulting in the best performance of flies. Results obtained from this study are very useful for the design and management of an efficient industrial black soldier fly mass rearing process.

## Introduction

The black soldier fly (BSF) *Hermetia illucens* (L.) (Diptera: Stratiomyidae) is a commonly distributed species which is gaining growing interest as a tool in the circular economy perspective for the capacity of its larvae to rapidly convert bio-waste into a protein-rich and fat-rich biomass, useful for many industrial purposes. Feeding on a wide variety of organic wastes of animal and vegetal origin [1–3] the BSF larvae reduce the waste amount up to 50–80% (wet weight) [4], transforming it in a residue suitable as soil amendment [5, 6]. The mature BSF larvae are valued for having a high content of protein (37 to 63% dry matter) and fat (7 to 39%

Regional Operational Program 2014-2020 [grant number PG/2015/737518] and by the Emilia Romagna region within the Rural Development Plan 2014-2020 Op. 16.1.01 – GO EIP-Agri - FA 5C, Pr. "BIOECO-FLIES". The funders had no role in study design, data collection and analysis, decision to publish, or preparation of the manuscript and that there was no additional external funding received for this study.

**Competing interests:** The authors have declared that no competing interests exist.

dry matter) [7, 8], with a great applicability in the feed and food industry [8–11], as well as for the production of biodiesel [12] and biodegradable bioplastics [13].

Nowadays research is focused on facing the challenges raised by indoor mass rearing of these insects. Although abundant studies were performed on the optimization of larvae rearing [3, 14–17], additional information on the adult breeding strategies must be deepened. Expanded knowledge on the individual reproductive potential of females, adult density and nutrition requirements [18, 19], may increase egg production and achieve a successful industrial process.

In the tropics, BSF adults mate and lay eggs yearlong, while in warm temperate regions natural breeding is restricted to a few generations [20, 21] when the temperatures are above 26°C [22, 23], as a consequence of the decrease of light duration and intensity due to seasonal variation [19, 22]. Factors such as light intensity, temperature and humidity play a decisive role in mating and oviposition performances of BSF flies [22–24]. BSF mating is considered to be dependent on direct sunlight [23, 25]; nevertheless, positive results were obtained by using artificial lighting such as quartz-iodine [26], LED emitting light [15, 16, 27], fluorescent tubes [15, 16], LED lamps and fluorescent tubes [27] or halogen lamps [16, 27], whereas rare earth lamps [26], Pro ultralight light system [23] or Sylvania Gro Lux system [23] proved to be ineffective. Oonincx et al. [15] performed specific tests on the physiological properties of the BSF visual system and suggested that a LED ratio of UV: B: G = 1: 1: 3 would optimize the visual pigment photo equilibrium and augment photoreceptor sensitivity.

The fecundity of females may be influenced by the nutritional content available during larval stages [3, 28] or by the type of nutrition administrated during adult stage [17, 18]. Moreover, a diet rich in proteins could increase the oviposition performances of BSF females [18]. Egg hatchability seemed to be positively influenced by LED lighting compared to fluorescent tubes [15] while a positive influence was observed by feeding adults with a combination of sugar, milk powder and bacteriological peptone, although the difference was not significant if compared to agar, water or no nutrition [18]. Although a comparison between adult life span under different light influence would be more complicated, due to different experimental conditions, focusing on nutrition has shown that the addition of sugar in adult diets increases longevity up to 3 times in males and 2 times in females [17, 27]. Based on the availability of nutrients in the natural habitat, studies suggested that diets rich in proteins or sugar improve the longevity of BSF adults [17, 18].

This study was designed to assess the simultaneous influence of three different light sources and three types of nutrition on the production parameters of BSF while focusing on adult performances. The experiment aimed at investigating the influence of the applied treatments on the life span of adults and on several egg production parameters such as: amount of eggs obtained, pre-oviposition period and oviposition period, egg laying performances of females, eggs/lux and egg hatching. Moreover, several measurements were performed to obtain data for an accurate estimation of the egg production (weight of a single egg, weight of an egg mass, number of eggs in one egg mass).

## Materials and methods

### Laboratory colony

The rearing of BSF was initiated in 2016 from larvae purchased from CIMI srl (Cuneo, Italy (CN)) and the colony continued to be maintained in the Laboratory of Applied Entomology—BIOGEST-SITEIA, Department of Life Science, University of Modena and Reggio Emilia, Reggio Emilia (RE), Italy.

Larvae were reared in glass containers (23 x 13 x 8 cm) at 27˚C and 70% relative humidity, with a 16:8 light-dark cycle [16, 27], using the standard Gainesville diet [29, 30] as feeding substrate, which was added twice a week. Pupae were manually separated and moved into cylindrical ventilated plastic containers for adult emergence. Three times per week the emerged adults were transferred to a transparent polyethylene container (50 x 40 x 25 cm) provided with 4 windows covered with a nylon net for ventilation. To ensure a greater longevity of adults, water and sugar cubes were provided and replaced twice a week [27]. A patent pending device (39), specifically developed in our laboratory, was used as the oviposition site. It consists of a cylindrical 3D printed case, containing the rearing substrate as attractant, surmounted by several removable plastic disks for eggs laying. The disks containing eggs were collected and placed directly on the rearing substrate without any other manual handling of eggs, three times per week.

## Experimental design

In order to assess the effects of light and diet on the oviposition traits and adult longevity, three artificial light sources (called mix of LED's, M, white LED, LD, and neon, NE) were tested simultaneously with three type of diets: sugar+water (SW), water (W) and nothing, (NO), resulting in 9 treatments: 1) Mix LED's + sugar+water (MSW); 2) Mix LED's + water (MW); 3) Mix LED's + nothing (MNO); 4) White LED + sugar+water (LDSW); 5) White LED + water (LDW); 6) White LED + nothing (LDNO); 7) Neon + sugar+water (NESW); 8) Neon + water (NEW); 9) Neon + nothing (NENO).

All the experiments were conducted between April 1$^{st}$–October 15$^{th}$ 2018, at 27˚C ± 0,5˚C and 70 ± 10% RH, with 16:8 (L:D) h photoperiod.

Description of light sources:

As shown in Fig 1, three different light sources were used during the experiments:

a. The mix of LEDs source consisted in an array of high-power light-emitting diodes (LEDs) with three different wavelengths: 365 nm (UV), 450 nm (Blue) and 515 nm (Green). The ratio of LEDs was 1:1:3, as suggested by Oonincx et al. [15]. The LEDs were mounted on a metal core PCB and attached to an aluminium thermal heatsink with two fans. The design of the PCB was optimized in order to homogenize the irradiation.

b. The Neon source was made with standard fluorescent tube lamps (Philips MASTER TL-D 36W 840, Made in France). The spectrum of such type of lamp is quite known and it consists in a well-defined characteristic wavelength related to the Neon gas emission lines. The Neon tube spectrum (available on the Philips website) contains several wavelengths: 360 nm, 400nm, 440 nm, 500nm, 560nm, 600nm and 640 nm.

c. The White LEDs source was a source made of high-power light-emitting diodes with a cold white spectrum (intensity 20.000 lux).

In order to compare the effect of different wavelengths, the different light sources were placed at a fixed distance from the cage during the experiment and the light intensity (lux) was measured (at the same point on the cage) for each configuration (S1 Fig–S3 Fig). The measured light intensity for each source was: 700 LUX for Mix of LEDs, 3700 LUX for Neon and 2450 LUX for white LEDs source.

The spectrum of each source of light were recorded between 360 and 940 nm by using a SPEKTRA-1 Kwant USB Spectrometer. The illuminance was measured with ILM 1332A Luxmeter, with a range from 0.01 lx to 200000 lux. Both spectra and illuminance were recorded in the geometric centre of the adult cage and could be assumed as the average values of the cage.

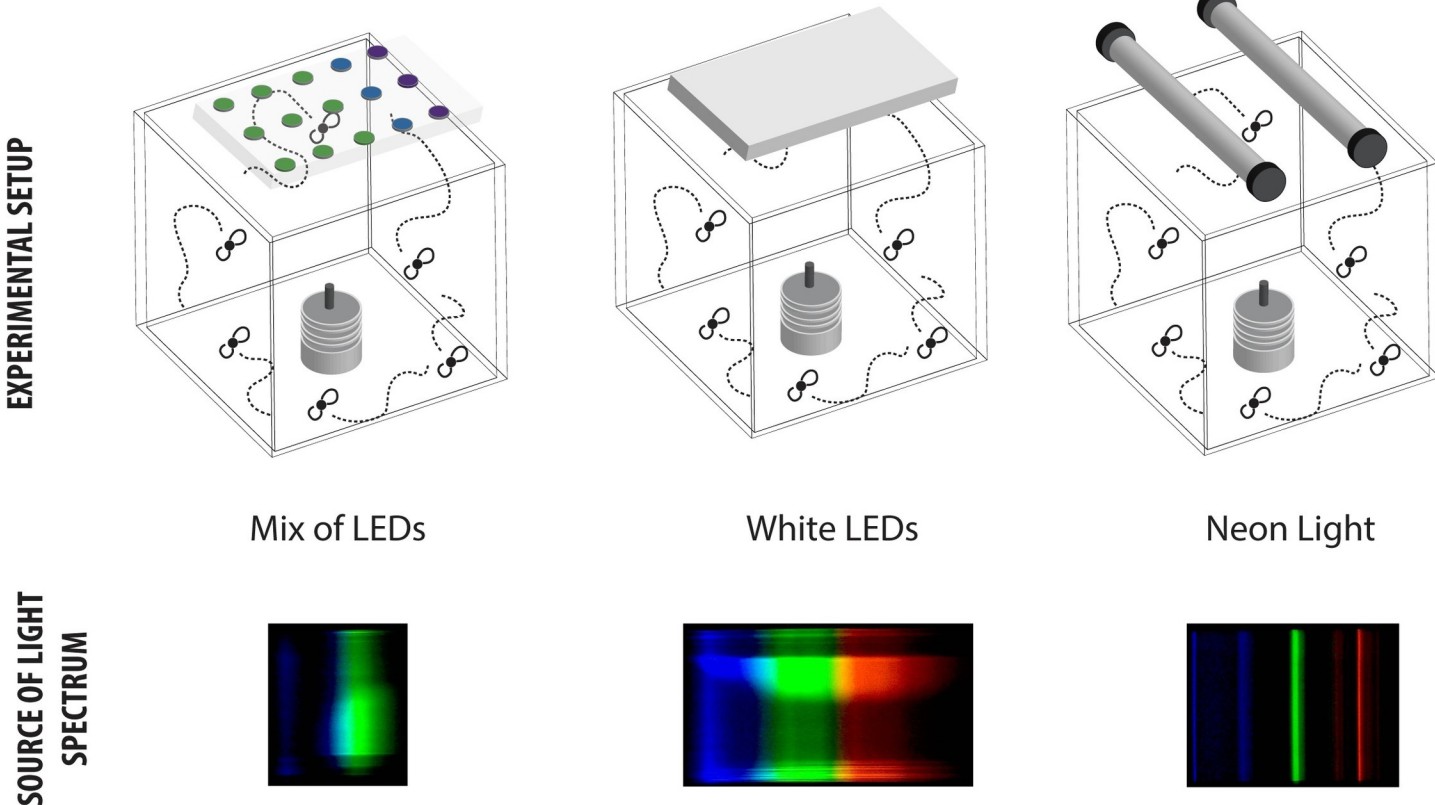

**Fig 1. Description of light sources.** Position of light sources (above); Spectra of light sources (below).

Description of the nutrition conditions tested for rearing the adults were:

1. SW treatment consisting in 2 white sugar cubes (4 g/sugar cube) and a plastic bottle cap (27 mm) filled with cotton soaked with water;

2. W treatment consisting in a bottle cap filled with cotton and water;

3. NO treatment with no sugar nor water was administrated during the trials.

Water and sugar cubes were refreshed if needed.

## Egg production parameters and adult life span

For each treatment, groups of newly emerged (<24 h) adults (15 females and 15 males) were released into wire-framed nylon cages (32.5 x 32.5 x 32.5 cm) purchased from BugDorm® company (United Kingdom). They were observed daily until the death of the last fly.

On a daily basis the egg masses (EM) were removed, counted and weighed with a KERN precision balance PLJ 700-3CM, readability 0.001. For egg collection, both the oviposition device as well the interior and corners of the cage were carefully checked. The identification of the different egg clutches was based on their slight colour differences and on the location on the oviposition device or inside the cage [27]. Dead flies were collected, sexed and their number was recorded daily.

Five replicates were performed for the treatments that included SW and W variables, whereas nine replicates were achieved for the treatments containing the variable NO.

The following parameters were considered and calculated:

a. Life span of adults = number of days between the flies' emergence and their death;

b. Overall egg amount (OEA) = total weight of the eggs (mg) laid by a group of adults during the oviposition period;

c. Minimum pre-oviposition period = number of days between the flies' emergence and the first EM was observed in the cage:

d. Oviposition period = number of days between the first and the last detection of an EM inside the cage;

e. Oviposition rate = Overall egg amount (mg) / number of days between fly emergence and the last EM observed;

f. Average weight of a single EM = Overall egg amount (mg) / Number of analysed EM;

g. Average number of eggs per one EM = Overall No. eggs / Number of analysed EM;

h. Average weight of a single egg = Overall weight EM (mg) / Overall number of eggs;

To obtain the parameters f-g-h, a total of 36 observations (S1 Table) were randomly performed in all the adult cages during the experimental period analysing a total of 391 egg masses. For each observation all the freshly laid (<24 h) egg masses were carefully removed, weighed with the KERN precision balance and the corresponding number of eggs was manually counted under a microscope (ZEISS, Stemi 2000-C) using a teasing needle.

Further, for a better evaluation, the data collected during each experimental period was categorized in six classes of observations based on their frequency, for each parameter of interest:

- Number of EM collected: observations containing a number of 0≥5 egg masses as first class (I), 5≥10 in second class (II), 10≥15 in third class (III), 15≥20 in fourth class (IV), 20≥25 in fifth class (V) and for a number higher than 25 egg masses in sixth class (VI);

- Weight of EM: observations containing between 0 and 0.067 g (I), between 0.067 and 0.133 (II), 0.133–0.200 (III), 0.200–0.267 (IV), 0.267–0.333 (V) and more than > 0.333 g of eggs (VI);

- Number of eggs: observations containing a number between 0 to 2500 eggs (I), 2500–5000 (II), 5000–7500 (III), 7500–10000 (IV), 10000–12500 (V), and more than 12500 eggs (> 12500- VI).

## Egg hatching

Considering that light source is a key factor for BSF mating and for obtaining fertilized eggs [23, 26] and that in previous literature no significant differences on egg hatchability were observed for different adult feeding strategies [18], we assessed the influence of different lights on egg hatching using a single diet treatment (W treatment).

The egg hatching parameter was calculated as the number of successfully hatched larvae compared to the estimated number expected [18] calculated considering the value of 0,028 mg weight of a single egg, previously obtained from the results of these trials. To determine this parameter, 5 subsequent experiments replicated 3 times were carried out (n = 15).

Newly emerged (<24 h) adults (15 females and 15 males) were released in a BugDorm® with a bottle cap filled with a cotton soaked in water. Between days 7–9 of the experiment [27], 10 mg of eggs (<24 h old) were collected, weighed on a Petri dish (90 mm diameter) and

placed directly on the rearing substrate, consisting in Gainesville House Fly diet [29, 30]. Controls were performed twice a week, when fresh substrate and tap water were added. After 14 days the larval biomass was removed and counted [15].

## Statistical analyses

All data analyses were done with various packages available in R Studio, ver. 1.1.414 (R Core Team, 2015)[31]. Correlations between eggs masses, their weight and the number of eggs were used to determine their relations and influence on each other. By using the "Hmisc" [32] package, ANOVA test was implemented as a descriptor of singular and combined influence of light and nutrition over the development of biological parameters. Significant interactions based on ANOVA were further analyzed by LSD test (both integrated in package "agricolae" [33]). A supplementary analysis was conducted on experimental factors and genders, in order to project in a 2-dimension PCA ordination the effect of light * nutrition.

Evaluation of the trade-offs between fly life span and their overall egg amount was performed with a non-metric multidimensional scaling (NMDS) ordination, available under "metaMDS" function. All ordinations were performed in "vegan" package [34].

## Results

### Life span of adults

The PCA analysis indicates a separation of treatments (Fig 2), especially due to the diet factor (p<0.001) (S2 Table). The 2D ordination graph explain 69.64% of the entire variation (Axis 1–51.78%; Axis 3–17.86%), which means that the extension of *light x nutrition* factors will be in horizontal plane. When the flies are not nurtured, a specific influence of light within each group (MN, LDN, NEN) can be noticed, inducing a lower variation. When water is combined

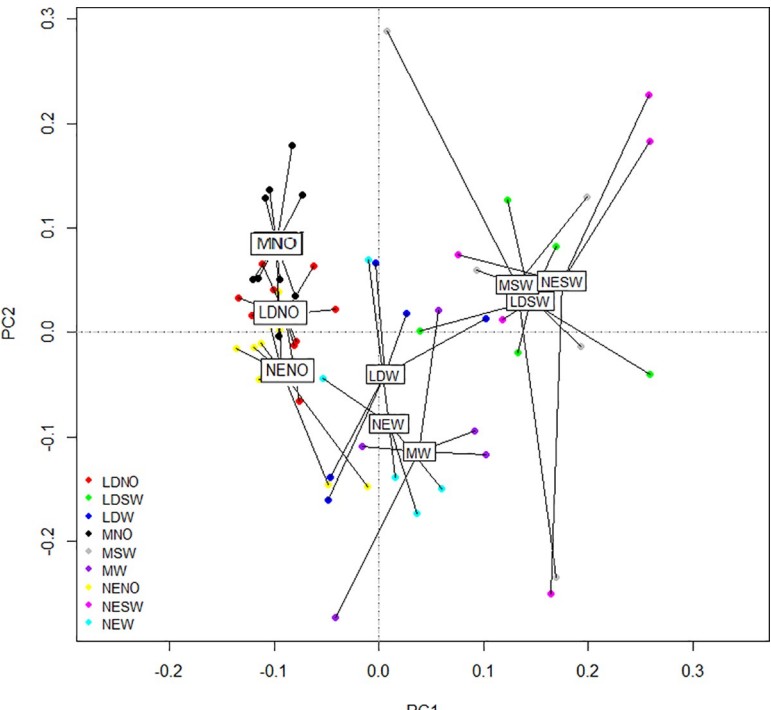

**Fig 2. PCA on complete reaction of adults to experimental factors, with separation caused by nutrition.**

with a light source, the internal variation increased. Although, the white LED (LDW) and the Neon (NEW) have different centroids on PCA, the variation between each group is similar, therefore the flies do not show a regular pattern in the presence of water. This aspect suggests a specific reaction of each individual to a source or water/food. The Mix of LEDs combined with water (MW), induces a high diversity on flies' life span, enhancing the individual response in a population of flies. By adding sugar in nutrition recipe, the lowest variation is observed in the case of White LED light (LDSW), which makes this variant fitted for longer life span observations/studies or as a control in the future test of different light sources. Sugar amplifies the effect of Mix of LEDs light and provide a powerful stimulus for BSF individuals, having position variations on the ordination, occupying the entire right part, above and below Axis 2. Overall, sugar and water move the centroids of the 3 light sources in the same spot on PCA which suggest that the mean response of each group to different treatments is similar.

Repeating the same analysis but considering gender as an experimental factor, an individual and specific reaction of individuals is shown (Fig 3), placed in different quadrants without any overlaps. Details on PCA factor loading are reported in S2 Table. Generally, females have a more homogeneous lifespan under NO treatment, whereas males tend to have similar longevity under W treatment, despite the light factor. Their orientation/position is in quadrants PC1 (+)/PC2 (+). The gradation is an indicator of a greater equilibrium of BSF female longevity, the vectors projected on the PCA axes being approximately all of a similar length. On the contrary, males show strong differences of lifespan, where the individual determinations (n = 855 males) provided a great support for this type of study.

In Figs 4 and 5 trade-off between life span of BSF flies (females and males) and OEA, is presented in a graphical way, under the influence of nutrition and light respectively. The stress observed is low (0.105), indicating a good bi-dimensional scaling.

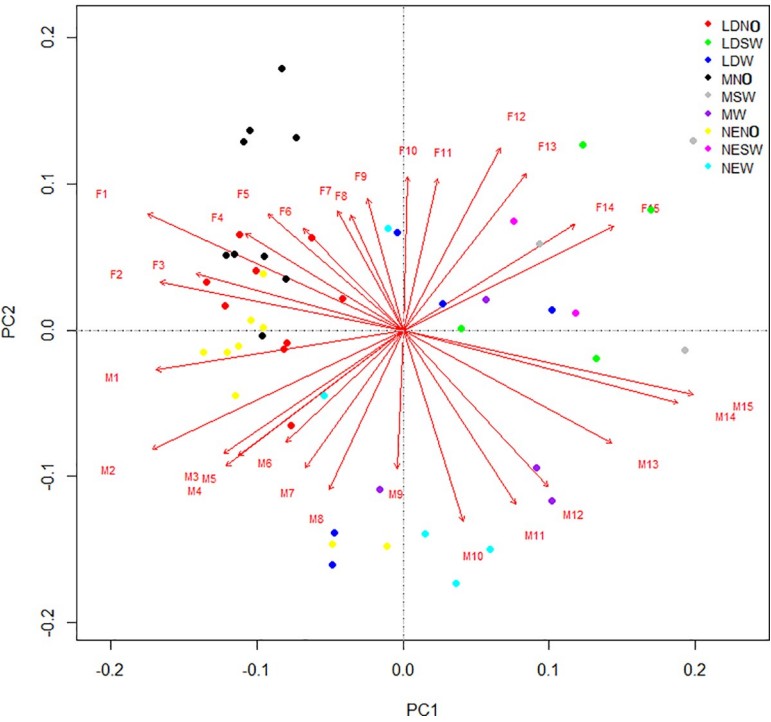

**Fig 3. PCA on complete reaction of adults to experimental factors, with separation caused by gender.**

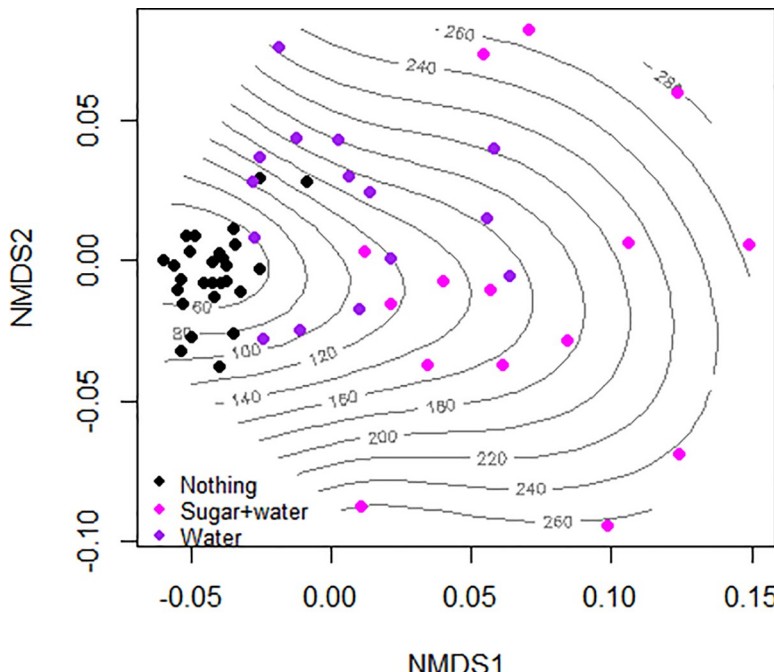

**Fig 4. Trade-off between the life span of adults (females and males) and the overall egg amount of eggs produced, under nutrition factor.** The lines represent the OEA (mg) produced per treatment and the life span of flies (days) are represented by dots.

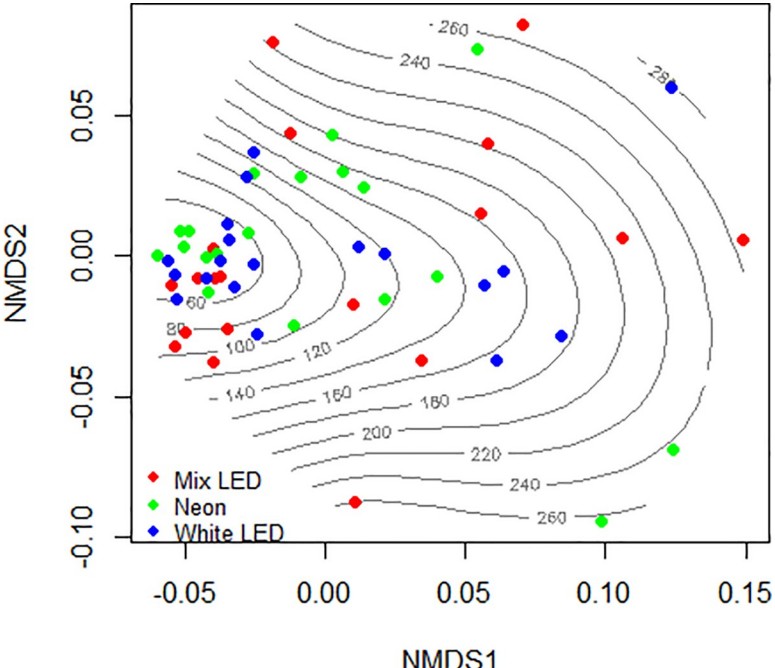

**Fig 5. Trade-off between the life span of adults (females and males) and the overall egg amount of eggs produced, under light factor.** The lines represent the OEA (mg) produced per treatment and the life span of flies (days) are represented by dots.

**Table 1. Life span of adults under different conditions of light and nutrition (Mean ± SD).**

| Treatments | Life span females (Days) | Total no. females | Life span males (Days) | Total no. males |
|---|---|---|---|---|
| MSW | 27.11 ± 8.71 a | 75 | 27.06 ± 13.74 a | 75 |
| NESW | 20.84 ± 3.74 a | 75 | 20.72 ± 4.98 ab | 75 |
| LDSW | 21.35 ± 4.71 a | 75 | 20.79 ± 3.22 ab | 75 |
| MW | 12.51 ± 1.81 b | 75 | 14.81 ± 3.03 bc | 75 |
| NEW | 13.49 ± 0.74 b | 75 | 15.16 ± 2.12 bc | 75 |
| LDW | 12.68 ± 2.08 b | 75 | 13.35 ± 3.32 bc | 75 |
| MNO | 8.38 ± 0.64 b | 135 | 7.19 ± 0.64 c | 135 |
| NENO | 7.91 ± 1.05 b | 135 | 8.02 ± 0.96 c | 135 |
| LDNO | 8.14 ± 0.59 b | 135 | 7.74 ± 0.61 c | 135 |

Different letters within each column are significantly different between treatments

By comparing the patterns of nutrition (Fig 4), the vast majority of eggs produced in the absence of nutrition is located in the interval 60–80 mg, with few cases where the flies produced more than 80–100 mg/cage. The addition of water increased the quantity of eggs produced from 100 up to 160 mg/cage, the large majority. The data is restricted in a square quadrant of NMDS ordination (Axis 1: -0.05 to +0.05 / Axis 2: -0.05 to +0.05). The mixed nutrition of flies, composed by sugar and water increased the egg production, having the majority of values between 140 and up to 280 mg.

The overall egg amount per cage is also influenced by the experimental light sources (Fig 5). Most of the flies exposed to the White LED produced less than 60–180 mg, followed by a similar pattern for Neon light. Whereas using the Mix of LEDs, it can be observed that the results are separated in two homogenous sections distributed as: half of flies produced less than 120 mg of eggs and the other half presented individual differences in the life span vs. egg production, within a range of 120–240 mg of eggs.

Females nurtured with SW lived a significantly longer number of days compared to W or NO diets (F = 29.96, p<0.001), while for males a significant difference was observed (F = 14.25, p<0.001) regarding all three levels of the diet factor (Table 1).

## Egg production parameters

Table 2 shows the correlation matrix beween the egg production parameters (OEA, number of EM, pre-oviposition and oviposition periods) and the tested experimental conditions (light and diet). The parameters i) overall egg amount, ii) number of EM, iii) pre-oviposition period and iv) oviposition period were correlated independently of the treatment factors applied. Nutrition factor has the highest influence, being positively correlated with each of the considered parameters. The light factor has a significant negative influence on the OEA, number of

**Table 2. Pearson correlation between the observed egg production parameters and the experimental factors.**

| | Light | Nutrition | Light * Nutrition | Tot. Number EM | Pre-oviposition | Oviposition |
|---|---|---|---|---|---|---|
| OEA (g) | -0.34 | **0.76** | **-0.26** | **0.98** | **0.4** | **0.82** |
| Tot. number EM | **-0.39** | **0.76** | **-0.31** | | **0.39** | **0.86** |
| Pre-oviposition period | 0.04 | **0.5** | 0.09 | | | **0.34** |
| Oviposition period | **-0.26** | **0.77** | -0.18 | | | |

Values marked in bold are significant at p < 0.05

OEA = overall egg amount, EM = egg mass

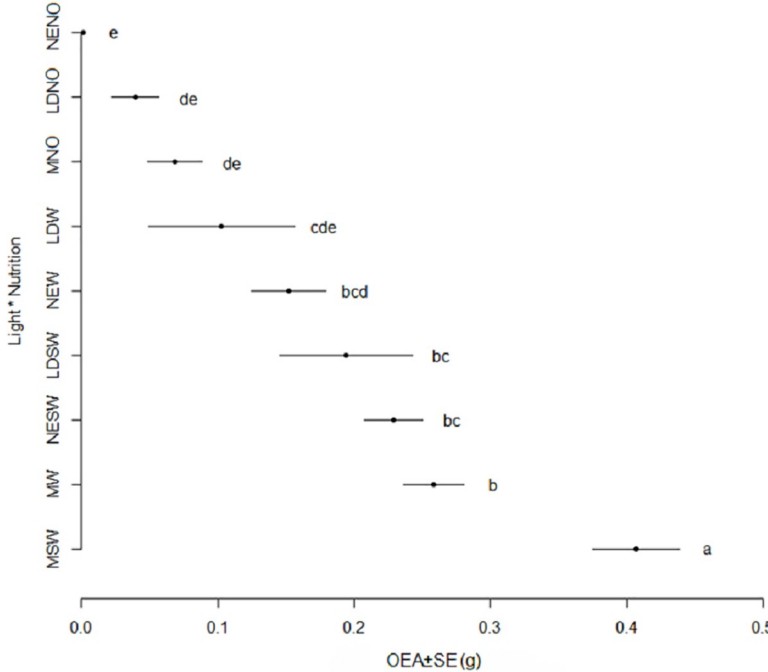

**Fig 6. Overall egg amount (g) obtained during the oviposition period (Mean ±SE).** Different letters indicate significant differences between treatments (p < 0.05).

EM and the oviposition period, although it slightly increased the preoviposition period. The interaction between light and nutrition reduces the impact of light, though maintains a significant correlation for the number (-0.31) and weight of EM (-0.26). Compared to oviposition period, the preoviposition period has a reduced influence on the number and weight of eggs, a fact further confirmed by the significant albeit reduced correlation between preoviposition and oviposition period (0.34).

The OEA and number of EM laid by BSF females among treatments differed significantly. The best performance was recorded for treatment MSW with 0.406 g of eggs and 37.2 EM respectively (Fig 6) and the worst in treatments NENO and LDNO with values between 0.001 g of eggs and 0.44 number of EM (Fig 7).

Overall, pre-oviposition period differed significantly and varied between 2 and 6.6 days (Fig 8). The longest pre-oviposition period was achieved in the treatments LDW (6.6 days) and NESW (6.4 days), with values significantly different compared to the other treatments; in the NENO treatment the shortest pre-oviposition period was recored (2 days). The other six treatments had intermediate values, not significantly different between each other, varying from from 3 to 5 days (S4 Fig).

Regarding the oviposition period, results were significantly different across treatments (S5 Fig), with the longest ones recorded in treatments containing SW (Fig 8), in particular for treatment MSW (31.8 days), although 80% of the eggs were mainly laid in the middle of the ovipostion period (Table 3). Flies with W or NO treatments laid eggs for shorter periods of time, with minor differences between the days needed to lay the egss and the necessary period to reach 80% of the OEA.

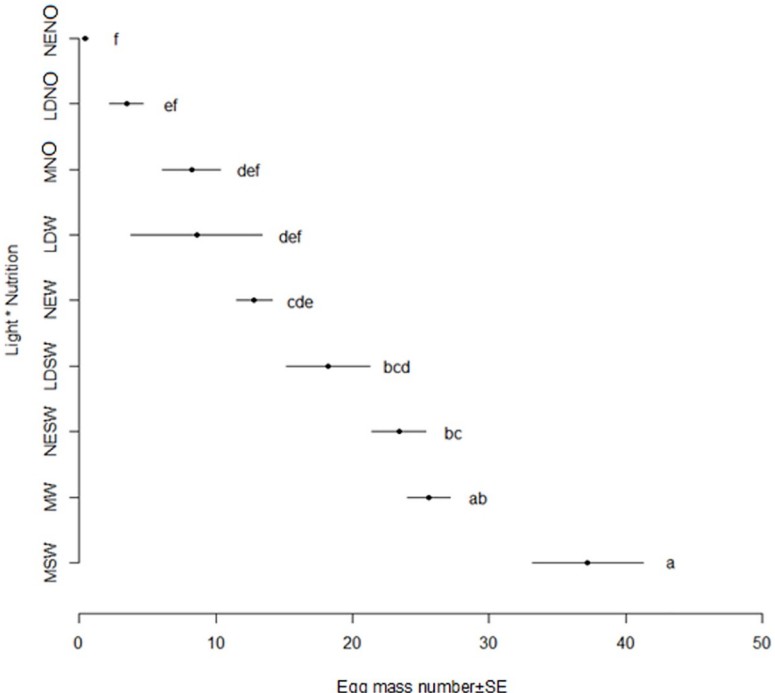

**Fig 7. Number of egg masses obtained during the oviposition period (Mean ±SE).** Different letters indicate significant differences between treatments (p < 0.05).

For treatments containing SW, the peak of oviposition was achieved in day 7, however considerable amounts of eggs were laid until day 22 of experiment (Fig 8). Regarding the treatments where the flies had only water, the majority of eggs were laid between days 5–15 after emergence. In absence of sugar and water, eggs were laid between days 5–9 of the experimental period (Fig 8).

Elements regarding female fecundity are presented in Table 3. Significant differences were observed for treatment MSW where the femailes laid the highest amounts of EM (2.28 EM/female) and the highest amount of eggs (27.10 mg eggs/female). Overall, female performances were stimulated by adding SW or only W. Higher amounts of eggs/day (oviposition rates) were obtained when 80% of OEA was reached if compared to the oviposition rate in reference to all oviposition periods, as the oviposition was rather concentrated on the first half of female's life.

Regardless of the type of nutrition, the highest amount of eggs was obtained in presence of mix of LEDs (Fig 9), that was also the source with the smaller intensity (700 LUX). The effect of the light source on eggs production is even more emphasized if we consider the overall quantity of eggs per LUX (OEA/LUX) for each light source; the Mix of LEDs resulted 9–10 times more effective than the other light sources.]

## Egg hatching success

Eggs successfully hatched when adults were exposed to all three type of lights (Table 4), with no significant differences among the treatments (F = 1,235, p = 0,301). The lowest variability was registered for mix of LEDs light treatment.

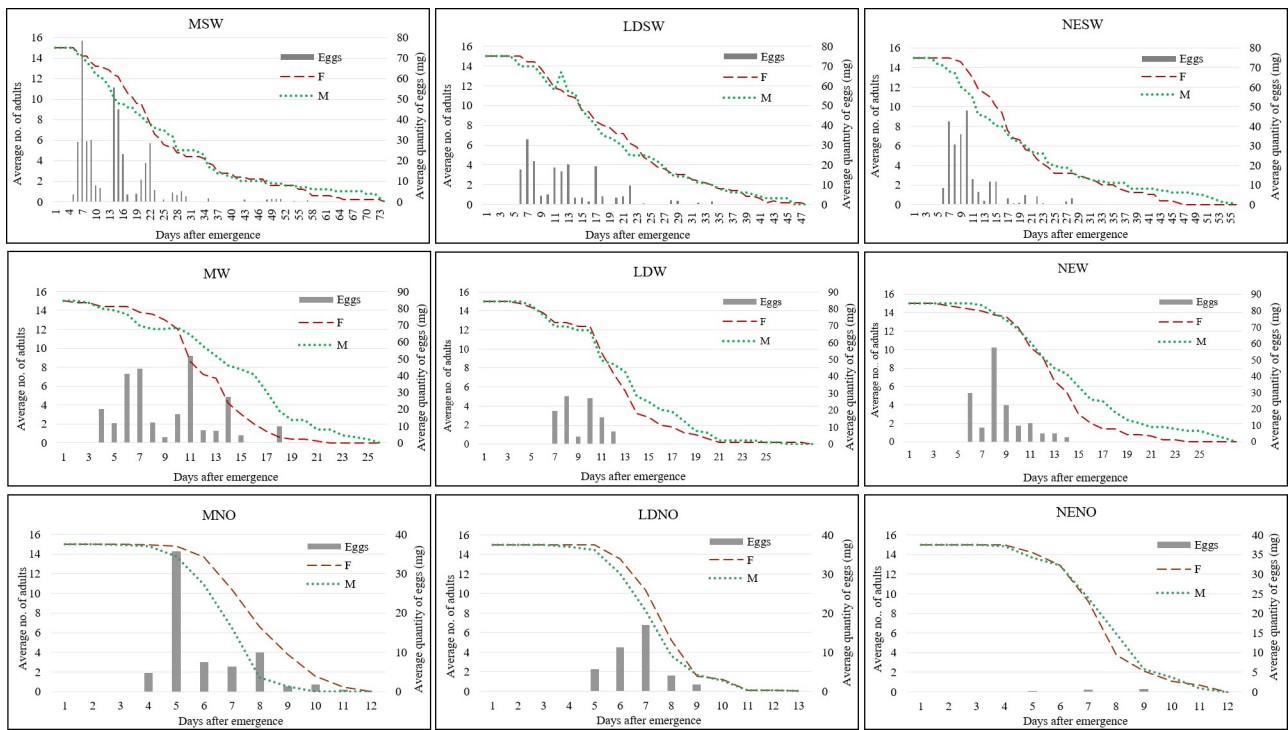

**Fig 8. Patterns of oviposition and survival curves of BSF adults observed in the different light and diet treatments.**

## Eggs and egg mass weight

The number of EM collected during the whole experiment varied from 2.77 to 30.5, with a majority of observations in classes I-III and averages close to the middle of intervals. The majority of observations made on the number of eggs belonging to class I and II (from 1000 to 3300 eggs) with averages close to the inferior limit of classes. Only 30% of the observations

**Table 3. Production parameters of BSF females (Mean ± SD).**

| Treatments | Eggs/female (mg) | EM/female | Tot. oviposition rate (mg/day) | Oviposition rate at 80% of eggs obtained (mg/day) | Period necessary to achieve 80% of the total egg amount (days) |
|---|---|---|---|---|---|
| MSW | 27.10 ± 4.72 a | 2.48 ± 0.60 a | 12.56 ± 4.65 ab | 19.71 ± 3.04 a | 18.60 ± 3.78 a |
| MW | 17.23 ± 3.31 b | 1.70 ± 0.23 ab | 16.87 ± 5.13 a | 19.71 ± 4.43 a | 11.80 ± 1.64 abc |
| MNO | 4.56 ± 4.01 de | 0.55 ± 0.42 def | 8.45 ± 6.48 abc | 10.78 ± 10.22 abc | 5.89 ± 2.93 cd |
| NESW | 15.27 ± 3.17 bc | 1.56 ± 0.29 bc | 9.64 ± 3.32 abc | 17.02 ± 3.38 ab | 11.40 ± 1.67 bc |
| NEW | 10.13 ± 4.03 bcd | 0.85 ± 0.19 cde | 13.43 ± 7.75 ab | 14.11 ± 7.58 ab | 10.20 ± 0.84 bc |
| NENO | 0.10 ± 0.17 e | 0.03 ± 0.05 f | 0.21 ± 0.33 c | 0.21 ± 0.33 c | 2.33 ± 3.64 d |
| LDSW | 12.93 ± 7.24 bc | 1.21 ± 0.45 bcd | 7.87 ± 3.15 abc | 11.46 ± 6.96 abc | 15.80 ± 3.83 ab |
| LDW | 6.85 ± 7.96 cde | 0.57 ± 0.72 def | 9.05 ± 10.70 abc | 9.37 ± 10.67 abc | 8.40 ± 4.83 cd |
| LDNO | 2.63 ± 3.44 de | 0.30 ± 0.25 ef | 5.15 ± 5.98 bc | 5.70 ± 7.37 bc | 3.67 ± 3.57 d |
| | *F = 22.94; p<0.001* | *F = 26.53; p<0.001* | *F = 4.817; p<0.001* | *F = 5.963; p<0.001* | *F = 17.63; p<0.001* |

Different letters within each column are significantly different between treatments (Fisher LSD test)

[x]Oviposition rate at 80% of eggs = 80% of the total amount of eggs per cage divided by the corresponding number of days (from emergence to the date when achieved 80% of egg amount)

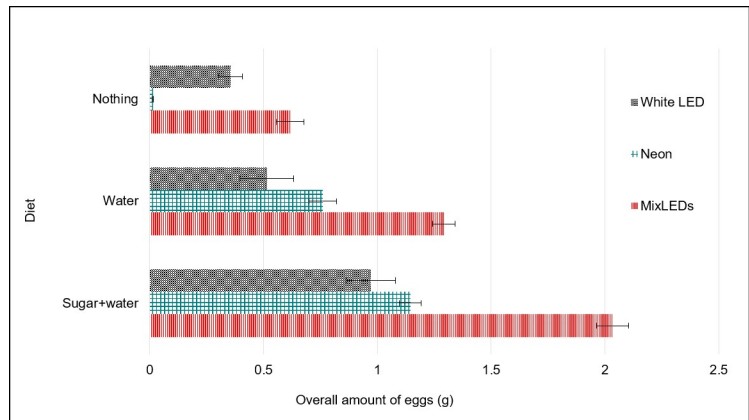

**Fig 9. Overall amount of eggs (g) per LUX obtained during the experimental period (Mean ± STDEV).**

were made on a number higher than 5000 eggs. The weight of EM follows the same trend, with average values that are close to the centre of the class interval.

As shown in Table 1, the correlation between the number of EM, their weight and the number of eggs in each EM was performed for each class (S3 Table). The measurements indicate a close linear positive relationship between the total number of eggs and their weight ($r = 0.99$) (Table 5). A positive correlation between the number of EM and the other two parameters, weight of an egg mass ($r = 0.96$) and number of eggs ($r = 0.96$), was also observed.

Concerning the first two classes, the numbers of eggs contained by one EM ranges between 407.66–414.36 eggs, while for the other classes varies between 462 and 493 eggs/EM. The average weight of an egg mass varied under 11.5 mg for inferior classes to above to 12.5 mg/EM for superior ones. Based on these data, the weight of a single egg was 0.028 mg, with values similar for 3 of the 6 classes analysed.

The data collected regarding the EM was also analysed, considering the principal factors applied (light, diet and light*diet). No positive correlations (Table 6) were observed between the number of EM, the weight of EM and the number of eggs per one EM collected, under the influence of experimental factors.

## Discussion

Rearing conditions have a strong influence on the egg production and BSF adult behaviour. The understanding of the influence of the single parameters and of their combinations is crucial for the development of an industrial mass rearing process. This work aimed at evaluating the effects of rearing BSF adults with different types of light and diet on the egg production parameters.

**Table 4. Percentage of larvae obtained from BSF females exposed to the three types of lights (Mean ± SD).**

|  | % of larvae hatched |
| --- | --- |
| MW | 53.23 ± 12.48 a |
| NEW | 43.15 ± 27.07 a |
| LEW | 55.73 ± 27.00 a |

Different letters within each column are significantly different among treatments

**Table 5. Correlations between the number of egg masses (EM), their weight and the number of eggs in each EM, based on the six classes identified.**

|  | Correlation | Class | | | | | |
|---|---|---|---|---|---|---|---|
|  |  | I | II | III | IV | V | VI |
| Number of eggs per one EM | 0.96 | 414.36 | 407.66 | 462.01 | 493.03 | 462.92 | 462.62 |
| Weight of an EM | 0.96 | 10.83 | 11.41 | 13.08 | 12.57 | 12.67 | 13.11 |
| Weight of a single egg | 0.99 | 0.026 | 0.028 | 0.028 | 0.025 | 0.027 | 0.028 |

According to our study, adult longevity is strongly influenced by the diet factor and also by the interaction between light and nutrition, where the latter could act in synergy. Both females and males showed longer life span (up to 27 days on average for both sexes) in the presence of sugar, while the combination with the Mix of LEDs (MSW) resulted in a high variation on the life span of the flies, being more suitable for studies on the BSF life span limitations.

The quantity of eggs is a good framework for non-metric multidimensional scaling (NMDS), highlighting the differences imposed by nutrition or light factors on flies. Based on the results we can suggest that the addition of water could be optimal for obtaining more stable results of life cycle assessment on flies. The addition of sugar produced more widespread results in the NMDS space, indicating an individual reaction of flies to nutrition that resulted in increased heterogeneity in flies' performance. Moreover, for each of the three light sources applied, we identified at least one case where 260–280 mg of eggs were produced, probably due to the individual potential of each fly.

The effect of nutrition on BSF adult performances was poorly investigated in the literature. According to Bertinetti et al. [18] egg production was higher when adults could feed on a combination of sugar, milk powder and bacteriological peptone, followed by a combination of agar, water and sugar, however no significant differences were observed between water nurtured and starved adults (exposed to neither food or water). According to our findings, the egg production parameters are influenced mainly by diet rather than light. The highest egg amount and number of egg masses were obtained when adults were in presence of both sugar and water, followed by only water whereas the lowest egg production was recorded in absence of water and sugar. This confirmed that presence of a food source positively affects egg production. Nevertheless, the greatest egg production was observed when adults were exposed to the mix of LEDs under all type of diet conditions, suggesting that the type of light has a secondary but still noticeable role in obtaining a high quantity of eggs. Previous studies did not observe significant differences in the number of egg masses obtained under different light sources: LED (UV:blue:green 1:1:1) vs. fluorescent lamps [15], green LED vs. halogen lamps [16], sunlight vs. quartz-iodine lamps [26], sunlight vs. fluorescent light supplemented with LED.

BSF are known as monogamous species that mate only once during their life span and according to Giunti et al. it is unlikely that they mate again later in life [35], whereas Samayoa et all. [36] suggested that the flies are able to mate again but can lay only a single egg mass. According to Lupi et al., the size of female ovaries were the highest 4 days after emergence and decreased until day 14 [17], with no influence from adult diet. After oviposition the females

**Table 6. Pearson correlations between the considered factors (light, diet and light\*diet) and the number of egg masses, weight of egg mass and the number of eggs in each egg mass (EM).**

|  | Light | Diet | Light * Diet |
|---|---|---|---|
| Number of eggs per one EM | -0.10 | -0.03 | -0.10 |
| Weight of an EM | -0.11 | 0.01 | -0.11 |
| Weight of a single egg | -0.07 | 0.01 | -0.07 |

die within a few hours [13] up to 4 days, with a maximum of 9 days post oviposition [36]. A high percentage (80–96%) of 4-day old females are able to mate [23, 35] and if provided with only water, they lay eggs two days after mating [23]. Similar results were obtained from our experiment in the presence of water only, with each different light source.

The oviposition period was significantly longer when sugar was added alongside water and, interestingly, differences were noticed according to the light source. Although 80% of the eggs were obtained between day 11–18 after the emergence of adults, the last egg was obtained 28, 34, 57 days after adult emergence in presence of neon lights, white LED, and Mix of LEDs respectively.

Our results indicate that the exposure of the adults to different types of lights did not have an effect on the percentage of eggs hatched, showing that on average about 43–56% of the initial number of eggs successfully hatched. Our data show the percentage of survival in day 14, while the highest mortality of larvae was reported for the first stage (51%) during the first four days [36]. The low number of larvae that survived might be due to the negative effects of fungi or other microbial growth as previously reported in literature [14, 36], also because in our case the fresh eggs (<24 h) were placed directly on the rearing substrate. In this work the aim was to test the egg hatchability after the flies were exposed to different lights and not to verify the best conditions to increase hatching. Therefore, in the view to improve egg hatching success for the implementation of industrial mass rearing further specific studies are needed.

By examining a high number of egg masses (391) obtained under different combinations of lights and diets, it emerged that the weight of a single egg mass was 10.83–13.11 mg and the number of eggs contained in one egg mass was 407.66–493.03, values that are similar to data reported in literature [16, 18, 37]. According to our findings, the weight of a single egg was 0.028 mg, with fluctuations at 0.025 mg, supporting previously reported data (0.028, 0.026 [18], 0.025 [21], 0.023–0.025 [37]) obtained with a much lower number of observations (10–75 egg masses). Although the egg masses were randomly collected by cages with flies exposed to different light and diet treatments, no positive correlations were obtained, therefore our results support the conclusions of Bertinetti et al. [38], that providing adults with a food source does not have any relevant effect on the size or weight of the egg masses.

In summary, according to our findings, the presence of sugar positively affects egg production and increases oviposition period and adult lifespan, suggesting that both females and males need energy for mating and egg laying. Light sources also affect egg production parameters, with the exposure to the Mix of LEDs resulting in increased egg laying, and also a longer oviposition period. It is known that the eyes of the BSF adult flies are sensitive to precise wavelengths and LED lights were recommended for BSF artificial rearing [15, 16].

Results obtained from this study are very useful for the design of an efficient industrial BSF mass rearing process. In an industrial perspective the optimal conditions of adult rearing are those that allow to obtain the highest egg production in the shortest period of time, therefore parameters such as the oviposition rate and the period necessary to achieve 80% of the total egg amount are extremely useful. By considering these parameters we can conclude that among the tested diet and light treatments the combination of the Mix of LEDs and presence of water is the best compromise.

## Supporting information

**S1 Fig. Light intensity of Neon light.**
(TIF)

**S2 Fig. Light intensity of Mix of LEDs light.**
(TIF)

**S3 Fig. Light intensity of White LED light.**
(TIF)

**S4 Fig. Preoviposition period observed during experimental period (Mean ±SE).** Different letters within each column are significantly different between treatments.
(TIF)

**S5 Fig. Oviposition period observed during experimental period (Mean ±SE).** Different letters within each column are significantly different between treatments.
(TIF)

**S1 Table. Number of observations perform for each experimental treatment.**
(TIF)

**S2 Table. Factor loading on PCA, based on reaction to experimental factors and gender.**
(TIF)

**S3 Table. Identified classes and number of observations for the three parameters fallowed: Number of egg masses, number of eggs and weight of egg masses (mg).**
(TIF)

## Acknowledgments

The authors are grateful to Joshua Gearing for English editing and to two anonymous reviewers for providing valuable suggestions that improved the quality of the manuscript.

## Author Contributions

**Conceptualization:** Laura I. Macavei, Giacomo Benassi.

**Data curation:** Giacomo Benassi.

**Formal analysis:** Laura I. Macavei, Vlad Stoian.

**Funding acquisition:** Lara Maistrello.

**Investigation:** Laura I. Macavei.

**Methodology:** Laura I. Macavei.

**Project administration:** Laura I. Macavei, Lara Maistrello.

**Supervision:** Lara Maistrello.

**Validation:** Laura I. Macavei.

**Visualization:** Laura I. Macavei, Giacomo Benassi, Vlad Stoian.

**Writing – original draft:** Laura I. Macavei.

**Writing – review & editing:** Laura I. Macavei, Giacomo Benassi, Lara Maistrello.

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
