## [Decision Letter · Decision Letter 0]

18 Feb 2020

PONE-D-19-36053

Optimization of Hermetia illucens (L.) egg laying under different nutrition and light conditions

PLOS ONE

Dear Dr. Macavei,

Thank you for submitting your manuscript to PLOS ONE. After careful consideration, we feel that it has merit but does not fully meet PLOS ONE’s publication criteria as it currently stands. Therefore, we invite you to submit a revised version of the manuscript that addresses the points raised during the review process.

1) The authors require to incorporate longevity in their analysis to ensure that you do not have a trade-off between longevity and reproduction,

2) The authors need to show data regarding egg hatchability and larval survival for the females treated with MSW.

We would appreciate receiving your revised manuscript by Apr 03 2020 11:59PM. To enhance the reproducibility of your results, we recommend that if applicable you deposit your laboratory protocols in protocols.io, where a protocol can be assigned its own identifier (DOI) such that it can be cited independently in the future. For instructions see: http://journals.plos.org/plosone/s/submission-guidelines#loc-laboratory-protocols

We look forward to receiving your revised manuscript.

Kind regards,

Humberto Lanz-Mendoza

Academic Editor

PLOS ONE

Additional Editor Comments (if provided):

The authors require to incorporate longevity in their analysis to ensure that you do not have a trade-off between longevity and reproduction, and the authors need to show data regarding egg hatchability and larval survival for the females treated with MSW.

Journal Requirements:

Reviewers' comments:

Reviewer's Responses to Questions

**Comments to the Author**

1. Is the manuscript technically sound, and do the data support the conclusions?

Reviewer #1: Yes

Reviewer #2: Yes

2. Has the statistical analysis been performed appropriately and rigorously? 

Reviewer #1: Yes

Reviewer #2: Yes

3. Have the authors made all data underlying the findings in their manuscript fully available?

Reviewer #1: Yes

Reviewer #2: Yes

4. Is the manuscript presented in an intelligible fashion and written in standard English?

Reviewer #1: Yes

Reviewer #2: Yes

5. Review Comments to the Author

Reviewer #1: Egg Hatchability is a reference to colony efficiency for a mass rearing industrial purposes, and so is the larvae survival. In my opinion in order to conclude that the nutrition and light treatment (MSW) is be the best treatment for industrial purposes, the authors need to show data regarding egg hatchability and larvae survival for the females treated with MSW. They only show data regarding the light but not the nutrition treatment.

The authors mention that egg hatchability is not influenced by adult nutrition, according to reference 28. However, the mentioned reference does show a positive impact of the nutritional treatment enhancing the hatching performance of H. illucens eggs. Therefore, it is necessary to present egg hatchability results regarding their MSW treatment, and the other treatments, along with the control, since it could be an interesting reference for optimizing rearing methods.

On the other hand, the percentage of hatchability seems to be low for H. illucens, and the authors do not show a reference nor control data.

Reviewer #2: Dear Authors and Editor. First at all my apologist for taking so long reviewing this paper. I was very. Very ill, and just now I take the time to read it.

The paper is intersecting. As a major comment are: a) I propose you to better present your results and b) incorporate longevity in your analyze just to ensure that you do not have trade-off between longevity and reproduction.

Line 27-30. Please be more specific here. Which gaps?

Line 95-97. Did you test all?

Results are very difficult to read. Maybe you can use a figure to explain it graphically. In addition, you may present the PCA first, and then, you can use ANOVAS with more interested results revealed in the PCA. That depends from you. My point is to you present much better your results.

Discussion, could you compare longevity between treatments. According to life-history theory, reproduction is traded of with longevity: The more eggs, the less longevity

6. PLOS authors have the option to publish the peer review history of their article (what does this mean?). If published, this will include your full peer review and any attached files.

Reviewer #1: No

Reviewer #2: No

---

## [Author Response · Author response to Decision Letter 0]

3 Apr 2020

Reviewer #1:

1) Egg Hatchability is a reference to colony efficiency for a mass rearing industrial purpose, and so is the larvae survival. In my opinion in order to conclude that the nutrition and light treatment (MSW) is be the best treatment for industrial purposes, the authors need to show data regarding egg hatchability and larvae survival for the females treated with MSW. They only show data regarding the light but not the nutrition treatment.

The authors mention that egg hatchability is not influenced by adult nutrition, according to reference 28. However, the mentioned reference does show a positive impact of the nutritional treatment enhancing the hatching performance of H. illucens eggs. Therefore, it is necessary to present egg hatchability results regarding their MSW treatment, and the other treatments, along with the control, since it could be an interesting reference for optimizing rearing methods.

On the other hand, the percentage of hatchability seems to be low for H. illucens, and the authors do not show a reference nor control data.

We totally agree that providing data on hatchability would have been an interesting reference for optimizing rearing methods. However, verifying the conditions to improve egg hatching was not the purpose of the present study. Certainly this will be the aim of our future research. We rewrote the paragraph and specified this in lines 425-430.

As well, we stated that the MSW treatment registered the highest amount of eggs produced. However, based on data provided by the oviposition rate (80% egg), we recommend the MW treatment for industrial purposes.

Besides, we rewrote many other parts to add specific useful information, as follows:

Line 68-71 we added and highlighted the information: ‘’Egg hatchability seemed to be positively influenced by LED lighting compared to fluorescent tubes (15)while a positive influence was observed by feeding adults with a combination of sugar, milk powder and bacteriological peptone, although the difference was not significant if compared to agar, water or no nutrition (18).”

Line 183-186 we added details: ‘’Considering that light source is a key factor for BSF mating and for obtaining fertilized eggs (23,26) and that no significant differences on egg hatchability have been observed for different adult feeding strategies (18), we assessed the influence of different lights on egg hatching with a single diet treatment (W treatment).”

Line 192-195 we added details: ‘’10 mg of eggs (<24 h old) were collected, weighed on a Petri dish (90 mm diameter) and placed directly on the rearing substrate, consisting in Gainesville House Fly diet (29,30). Controls were performed twice a week, when fresh substrate and tap water were added. After 14 days the larval biomass was removed and counted (15)’’.

Lines 425-430 we added explanation: “The low number of larvae that survived might be due to the negative effects of fungi or other microbial growth as previously reported in literature (14,38), also because in our case the fresh eggs (<24 h) were placed directly on the rearing substrate. In this work the aim was to test the egg hatchability after the flies were exposed to different lights and not to verify the best conditions to increase hatching. Therefore, in the view to improve egg hatching success for the implementation of industrial mass rearing further specific studies are needed.’’ 

Reviewer #2:

Dear Authors and Editor. First at all my apologist for taking so long reviewing this paper. I was very. Very ill, and just now I take the time to read it.

The paper is intersecting. As a major comment are: 

1) a) I propose you to better present your results 

Thank you very much for your suggestion, we reorganized all the ‘’Material and methods’’, ’’Results’’ and ‘’Discussion’’ sections. The article now has a better flow and the results are easier to follow. 

2) b) incorporate longevity in your analyze just to ensure that you do not have trade-off between longevity and reproduction.

Due to the fact that the overall egg amount and the life span or BSF females and males were already individually analyzed and presented, we introduced a Non-metric analyze for the trade-off between longevity and reproduction of BSF flies, under the influence of nutrition and light, in a graphical way (Fig 4 - 5). A whole paragraph has therefore been added to ‘’Results’’ (lines 244-266).

3) Line 27-30. Please be more specific here. Which gaps?

The sentence has been modified and several issues/elements were added.

Currently lines 51-54: ‘’Although abundant studies were performed on the optimization of larvae rearing (3,14–17), while additional information on the adults breeding strategies, such as individual reproductive potential of females or adult density and nutrition needs must be deepened (18,19), in order to increase egg production and achieve a successful industrial process.’’

4) Line 95-97. Did you test all? 

After the changes made, the lines correspond to the numbers 119-122. 

Neon tube is the common name of fluorescent lamp. The emission of these type of light source is due to the excitation of different atoms and, therefore, the spectrum emission consists in a series of several specific wavelength or lines (as we measured with a spectrometer).

We did not try different fluorescent light because their spectrum is always due to the same atoms: mercury, terbium and europium. Different mixes of these atoms can produce on the human eye the effect of different colors. In reality, the characteristic wavelengths are the same, and, the color effect is due to the different relative intensity of the atoms.

For the purpose of our experiment we are interested only in the effect of different wavelengths on the BSF, for this reason we can conclude that the effect of the lamp, used in the article, can be representative for all the fluorescent lamp family. 

5) Results are very difficult to read. Maybe you can use a figure to explain it graphically. In addition, you may present the PCA first, and then, you can use ANOVAS with more interested results revealed in the PCA. That depends from you. My point is to you present much better your results

We began the ‘’Results’’ presentation by exposing the Life Span of adults and we improved the interpretation of PCA analysis (lines 213-227). Further, starting with Line 244, we continued with the presentation of Trade-off analysis.

6) Discussion, could you compare longevity between treatments. According to life-history theory, reproduction is traded of with longevity: The more eggs, the less longevity.

 Please see previous comments at point 2)-b) and we added more discussions in lines 384-395.

---

## [Decision Letter · Decision Letter 1]

8 Apr 2020

Optimization of Hermetia illucens (L.) egg laying under different nutrition and light conditions

PONE-D-19-36053R1

Dear Dr. Macavei,

We are pleased to inform you that your manuscript has been judged scientifically suitable for publication and will be formally accepted for publication once it complies with all outstanding technical requirements.

With kind regards,

Humberto Lanz-Mendoza

Academic Editor

PLOS ONE

Additional Editor Comments (optional):

Reviewers' comments:

Reviewer's Responses to Questions

**Comments to the Author**

1. If the authors have adequately addressed your comments raised in a previous round of review and you feel that this manuscript is now acceptable for publication, you may indicate that here to bypass the “Comments to the Author” section, enter your conflict of interest statement in the “Confidential to Editor” section, and submit your "Accept" recommendation.

Reviewer #1: All comments have been addressed

Reviewer #2: All comments have been addressed

2. Is the manuscript technically sound, and do the data support the conclusions?

Reviewer #1: Yes

Reviewer #2: Yes

3. Has the statistical analysis been performed appropriately and rigorously? 

Reviewer #1: Yes

Reviewer #2: Yes

4. Have the authors made all data underlying the findings in their manuscript fully available?

Reviewer #1: Yes

Reviewer #2: Yes

5. Is the manuscript presented in an intelligible fashion and written in standard English?

Reviewer #1: Yes

Reviewer #2: Yes

6. Review Comments to the Author

Reviewer #1: In my opinion the manscrit is ready for publication, all the comments were addressed. The insect industry is a new

fast growing insdutry, and this kind of research is needed.

Reviewer #2: Dear authors,

Thank for your effort. All comments were responded and the paper is interesting. I am happy to accept this paper.

7. PLOS authors have the option to publish the peer review history of their article (what does this mean?). If published, this will include your full peer review and any attached files.

Reviewer #1: No

Reviewer #2: No

---

## [Editor Report · Acceptance letter]

13 Apr 2020

PONE-D-19-36053R1 

Optimization of *Hermetia illucens* (L.) egg laying under different nutrition and light conditions 

Dear Dr. Macavei:

I am pleased to inform you that your manuscript has been deemed suitable for publication in PLOS ONE. Congratulations! Your manuscript is now with our production department. 

With kind regards,

on behalf of

Dr. Humberto Lanz-Mendoza 

Academic Editor

PLOS ONE